# SiC Doping Impact during Conducting AFM under Ambient Atmosphere

**DOI:** 10.3390/ma16155401

**Published:** 2023-08-01

**Authors:** Christina Villeneuve-Faure, Abdelhaq Boumaarouf, Vishal Shah, Peter M. Gammon, Ulrike Lüders, Rosine Coq Germanicus

**Affiliations:** 1LAPLACE (Laboratoire Plasma et Conversion d’Energie), Université de Toulouse, CNRS, UPS, INPT, 118 Route de Narbonne, CEDEX 9, 31062 Toulouse, France; christina.villeneuve@laplace.univ-tlse.fr; 2CRISMAT UMR6508 (Laboratoire de Cristallographie et Sciences des Matériaux), Normandie University, Ensicaen, Unicaen, CNRS, 14000 Caen, France; abdelhaq.boumaarouf@ensicaen.fr (A.B.); ulrike.luders@ensicaen.fr (U.L.); 3School of Engineering, University of Warwick, Coventry CV4 7AL, UK; vishal.shah@warwick.ac.uk (V.S.); p.m.gammon@warwick.ac.uk (P.M.G.)

**Keywords:** local oxidation, atomic force microscopy, C-AFM, doping, successive AFM characterizations, doped silicon carbide, MOSFET

## Abstract

The characterization of silicon carbide (SiC) by specific electrical atomic force microscopy (AFM) modes is highly appreciated for revealing its structure and properties at a nanoscale. However, during the conductive AFM (C-AFM) measurements, the strong electric field that builds up around and below the AFM conductive tip in ambient atmosphere may lead to a direct anodic oxidation of the SiC surface due to the formation of a water nanomeniscus. In this paper, the underlying effects of the anodization are experimentally investigated for SiC multilayers with different doping levels by studying gradual SiC epitaxial-doped layers with nitrogen (N) from 5 × 10^17^ to 10^19^ at/cm^3^. The presence of the water nanomeniscus is probed by the AFM and analyzed with the force–distance curve when a negative bias is applied to the AFM tip. From the water meniscus breakup distance measured without and with polarization, the water meniscus volume is increased by a factor of three under polarization. AFM experimental results are supported by electrostatic modeling to study oxide growth. By taking into account the presence of the water nanomeniscus, the surface oxide layer and the SiC doping level, a 2D-axisymmetric finite element model is developed to calculate the electric field distribution nearby the tip contact and the current distributions at the nanocontact. The results demonstrate that the anodization occurred for the conductive regime in which the current depends strongly to the doping; its threshold value is 7 × 10^18^ at/cm^3^ for anodization. Finally, the characterization of a classical planar SiC-MOSFET by C-AFM is examined. Results reveal the local oxidation mechanism of the SiC material at the surface of the MOSFET structure. AFM topographies after successive C-AFM measurements show that the local oxide created by anodization is located on both sides of the MOS channel; these areas are the locations of the highly n-type-doped zones. A selective wet chemical etching confirms that the oxide induced by local anodic oxidation is a SiOCH layer.

## 1. Introduction

Recently, silicon carbide (SiC) has become a mature wide-band-gap semiconductor in microelectronics networks and especially in electric-power conversion [1] and has shown continuous high-temperature integrated circuit operation over 800 °C [2]. It has demonstrated exceptional characteristics compared to silicon, with numerous advantages in high-temperature and high-frequency applications due to its high-critical electric field and its high thermal conductivity [3]. SiC devices have already demonstrated their excellent performances in terms of energy conversion and system losses, with increased devices density [4]. SiC MOSFETs have shown their interest for power networks, especially concerning the resistance to high voltages and currents [5].

The high integration density for small scale devices, the process requirements, but also the reliability analysis require precise and local techniques to analyze devices at the SiC-die level. The details of the SiC inside the die, i.e., the achieved dopant level and other fundamental properties, as well as the interactions at its interfaces with other materials are important to be analyzed to understand the macroscopic performances of the full device, compared to its modelled attributes. Based on atomic force microscopy (AFM), several modes are becoming essential techniques to probe and examine local electrical properties at the nanoscale. These modes include modes based on conduction-like scanning capacitance microscopy (SCM) [6,7], scanning non-linear dielectric microscopy (SNDM) [8,9,10], conductive AFM (C-AFM) [11,12,13,14,15] or scanning spreading resistance microscopy (SSRM) [16,17,18,19], and a mode based on microwave interaction sMIM (scanning microwave impedance microscopy) [20,21]. Nowadays, all these modes are used extensively for their capability in terms of spatial resolution, detection, analysis of doping species, conductivity, resistivity or isolation properties. During the measurements with these modes, an electrical voltage is applied on the tip scanning the surface of the sample, with a mechanical contact between the tip and the surface. When the measurements are carried out under ambient atmosphere, i.e., air containing moisture, the formation of a water meniscus surrounding the tip occurs due to the adsorption of the humidity. This water bridge changes the local electrochemical behavior around the AFM tip and can cause a localized oxidation of the material under investigation [22]. For silicon, several extended studies have been conducted since the 1990s. At the tip–sample nano-contact, a local electrochemical cell is created (Figure 1), where the AFM silicon tip plays the role of a cathode. When a negative appropriate bias is applied to the tip, the high electrical field (in the range of 10^9^–10^10^ V∙m^−1^) in the electrochemical nano-cell induces the creation of O^−^, OH^−^, and H^−^ ions. The polarization applied on AFM favors water electro dissociation. With the molecular transport (ionic-diffusion mechanisms) through the water meniscus, negative ions flow towards the surface and recombine with holes (h+) [23] at the surface of the silicon. Therefore, this anodic oxidation mechanism allows for a direct oxide growth located under the AFM tip.

This mechanism is called AFM local anodic oxidation (AFM-LAO). AFM-LAO is used to create and manage oxidized nano-structures like nano-dots or lines also called (nano-patterning). The grown oxide thickness depends on several parameters such as the silicon properties and scanned AFM parameters [24].

For SiC, authors have reported such a local oxidation by a polarized AFM tip during the scan [25,26]. As for the silicon, the main origin of this phenomenon is due to the ionization of the water meniscus presence under electric fields of at least 8 × 10^6^ V/m [25,27,28]. The AFM-LAO oxide growth rate for SiC was studied as a function of the applied DC bias [29,30], bias pulses [30], loading force, scan rate and/or doping level for the 6H-SiC and 4H-SiC [31]. In [31], authors also show that the AFM local oxidation can be promoted by high doping concentrations; in this case, two doping levels of phosphorus-implanted 4H-SiC are experimentally compared.

The growth mechanisms during this local anodization and, more particularly, the influence of the SiC doping have not been comprehensively investigated up to now. In this context, this study aims to investigate the influence of the SiC doping level on this growth mechanism during conductive AFM scans. This study goes a step further considering two aspects: (i) the fine quantification of the local oxidation as a function of the doping level with the study of a gradual SiC epitaxial-doped layers with nitrogen (N); and (ii) the study of the local oxidation mechanism after consecutive scans of the cross-section of a power SiC MOSFET. To reach this goal, a staircase doping sample is scanned and analyzed. The experimental results provided by AFM are combined with finite element modeling, to evaluate the electric field and current distributions when the tip is in contact with the sample surface. Results are extended to a classical SiC MOSFET by studying the evolution of the surface cross-section after successive C-AFM scans.

## 2. Materials and Methods

### 2.1. Description of the SiC Staircase Sample and MOSFET Device

Two kinds of samples were used (Figure 2): epitaxial SiC layers with various doping levels, and a commercial off-the-shelf (COTS) SiC MOSFET. The epitaxial SiC (sample A) was specially designed for this study; the sample is a staircase SiC multilayer doped in situ with nitrogen (N) with dopant concentrations from 5 × 10^17^ to 10^19^ at/cm^3^. This sample is fabricated to determine the threshold of the doping level for the LAO mechanism. Epitaxial layers were grown at Warwick on 100 mm diameter, 4° off-axis, n+ 4H-SiC wafers using an LPE ACiS M8 chemical vapor deposition (CVD) reactor. The 6 µm thick 4H-SiC homoepitaxial layers were grown using trichlorosilane (HCl_3_Si, TCS) and ethylene (C_2_H_4_) at a C/Si ratio of 0.46 with nominal growth rates of 5 µm/hr at a temperature of 1650 °C, where N_2_ was used to dope the layers. An optical view of the sample A is shown in Figure 2; note that another SiC wafer is used to avoid edge effects during the polishing state. Sample B is a classical TO-molded packaged COTS power SiC MOSFET that has a planar structure (third generation) with a vertical double implantation. The nominal voltage and current rating are 900 V and 23 A, respectively, and R_DS_ = 12 Ω. For AFM scans, a cross-sectional sample was prepared, keeping the electrical access through the TO packaging for electrical biasing during the AFM scans [32]. The axis of the cut is perpendicularly to the gate contacts. The surface of the cross-sectional sample was polished with lapping films. Decreasing granularity disks were used to obtain a very low roughness, i.e., a mirror-like surface. The schematic cross-section SiC epilayers represent the multilayer structure deposited on the 4H-SiC substrate. A confocal view (Figure 2) reveals the multigate structure of the device. A schematic of the MOSFET structure is also reported.

### 2.2. Atomic Force Measurement Details

For the AFM measurements, a Dimension Icon AFM from Bruker is used. The topography and the local conductivity were simultaneously recorded in the C-AFM [7] mode. A conductive platinum–iridium (PtIr) coated tip (SCM-PIC, Bruker, Billerica, MA, USA) was used with a tip radius around 25 nm. The measured spring constant of the used cantilever was determined to be about 0.1 N/m. This low spring constant assures a good electrical contact with the oxide at the surface of the cross-sectional sample without indenting the thin oxide layer. The image resolution was 512 × 512 pixels with a slow scan rate of about 0.3 Hz during the electrical measurements. AFM acquisitions were performed under ambient conditions at room temperature with a relative humidity of 45%.

### 2.3. Simulations of the Electric Field and Current Distribution under the Tip–SiC Nanocontact

In ambient conditions, the electric field located around the AFM tip and the presence of adsorbed water are the origin of the local oxidation of the scanned surface [33,34]. In fact, during AFM-LAO, by applying a negative bias to the AFM tip, a local electrochemical cell was formed around the tip and could allow local anodic oxidation (Figure 1). In order to determine the electric field distribution in the nano-system, the water meniscus/oxide/SiC region, an electrostatic model, was deployed. A 2D-axisymmetric finite element model was developed with COMSOL Multiphysics. The AFM tip was computed as a classical truncated cone with 10 µm height and 14° aperture angle, ending with a semi-spherical apex (curvature radius R_c_ = 25 nm). In our model, the water meniscus was simulated with a height of 30 nm and a radius of 50 nm at the interface with the surface sample. The rest of the tip was supposed to be surrounded by air, modeled by a box whose dimensions are large enough to avoid edge effects [35]. The SiC layer sample is modeled by a 500 nm thick material covered by an oxide layer (representing the LAO oxide) whose thickness ranged from 0 nm to 6 nm. As the presence of carbon cannot be excluded, a SiOCH [30,36] layer was simulated as the oxide layer.

The local electric field $\vec{E}$ derives from the potential *V*:(1)$$\vec{E}=-\vec{grad}\;\left(V\right)$$

Moreover, to simulate the local electric field distributions $\vec{E}$, two approaches were developed. First, to investigate the influence of water bridge presence and SiOCH layer thickness on electric field distribution, the Poisson’s equation was solved in the surrounding air, in the water meniscus and in the device.
(2)$$div\left(\varepsilon \vec{E}\right)=\rho$$
where ρ is the charge density and *ε* the dielectric permittivity.

The relative dielectric permittivity of water, SiC and SiOCH are 80.0, 9.2 and 3.0, respectively.

For the classical simulations, typical boundary conditions were applied; no charge conditions (i.e., zero potential) were applied on the free boundaries of the simulation box to avoid edge effects. Moreover, the water meniscus, SiOCH and SiC were considered free of charge (i.e., ρ = 0 C cm^−3^).

Secondly, to investigate the influence of SiC doping on the current, the following equation was solved:(3)$$\vec{J}=\sigma \vec{E}$$
with σ the conductivity of the different materials.

In the water bridge, the conductivity is considered to be constant (5 × 10^6^ S/m) and variable for the SiC and SiOCH layers. For the SiOCH layer, a conductivity ranging from 1.7 × 10^−5^ S/m (Si in excess) to 1 S/m (defect or impurities) is considered to reproduce the poor insulating properties of this oxide. For the SiC layer, a conductivity ranging from 1.7 × 10^3^ S/m (i.e., doping concentration of 10^17^ at/cm^3^) to 2 × 10^4^ S/m (i.e., doping concentration of 5 × 10^9^ at/cm^3^) was used.

## 3. Results and Discussion

### 3.1. The Tip–Sample Interaction in Ambient Conditions: Signature of the Water Meniscus

In AFM, the force–distance curves (FDCs) provide quantitative acquisitions of forces acting between the AFM tip and studied sample, at a nanometer scale. FDCs allow for the probing of the forces inherent to the tip–sample interaction: Van der Waals, electrostatic, adhesion or mechanical forces [37,38]. More particularly, when the tip retracts from the surface, the adhesion signal between the AFM tip and surface represents the superposition of forces due to the electrostatic force F_el_, the Van der Waals force F_VdW_, the capillary forces F_cap_ and chemical bonds or acid–base interactions F_chem_. F_cap_ depends on the relative humidity and the tip/sample hydrophobicity, and F_el_ could modify the water meniscus shape around the tip.

In spectroscopy mode (acquisition on a unique pixel without scanning), AFM FDC were recorded. During the tip–sample approach and retract movements, by measuring the AFM cantilever deflection, the force F follows this relation:(4)F=kSd
with *k* the experimental spring constant of the cantilever, *S* the sensitivity of the AFM cantilever and *d* is the laser deflection induced by AFM cantilever bending.

Experimental results are shown in Figure 3: the AFM apex interacts with the surface of the SiC in ambient conditions, with and without negative bias. Approach and retract curves are reported. These experiments were done for the SiC epitaxial layer with a doping level of 4 × 10^17^ $\mathrm{at}/{\mathrm{cm}}^{3}$. Without polarization, far from the SiC surface, the force applied to the AFM cantilever is zero. During the approach, when the tip is close to the surface, the Van der Waals attractive force occurs. The following linear relationship between force and Z position is characteristic of the mechanical properties of the sample surface. In a second step, when the tip is retracted from the surface, the adhesion force increases the spring constant of the cantilever and a supplementary force is needed to pull off the tip from the surface. For our experiment without bias (Figure 3a), a significant adhesion force of 1.9 nN was measured. As the Van der Waals force is low (not observed in the approach curve), the adhesion force is mainly related to capillary condensation (F_chem_ is lower than F_VdW_). When a negative DC bias voltage of −10 V is applied on the tip (sample back-side is grounded), a modification of the FDC is observed (Figure 3b). During the tip approach, a curvature of the FDC upon contact is detected. This attractive force corresponds to the long-range electrostatic force. Indeed, an attractive electrostatic force of around 0.8 nN was measured at the contact point. During the retract movement, the adhesion force needed for pull-off increases to 3 nN (compared to the curve without polarization), which corresponds to an enhancement factor close to 1.5. This AFM measurement proves that a nanomeniscus is present around the tip, leading to a supplementary, strong adhesive force [39,40]. When a negative V_DC_ is applied to the AFM probe, the generated local electric field induces the growth of the water meniscus located around the tip. The FDC modification under electric bias is related to the increase in the electrostatic force and interaction with the liquid meniscus formed by capillary condensation at the tip–sample contact around the AFM tip. Schematics of the water bridge (capillary condensation) around the conductive AFM tip are shown in Figure 3. From the water meniscus breakup distance measured without and with polarization, the water meniscus volume is increased by a factor of three under polarization [41].

The local oxidation growth mechanism of metals, described by the Mott–Cabrera model [42], is due to the existence of the water nanomeniscus with a high local electric field. In fact, the local water nanomeniscus bridges the electrical conduction mechanism and provides the anions to permit the chemical oxidation by driving hydroxyl anions towards the surface of the sample. In order to determine the electric field distribution in the near environment of the conductive AFM tip during the contact with the 4H-SiC surface, an FEM model was developed. The electric field was computed using Equations (1) and (2) and taking into account the dielectric permittivity of water, air, and the SiOCH and SiC layers. Figure 4a represents the electric field distribution when the tip is in direct contact with the SiC surface (i.e., no SiOCH layer) and without a water meniscus. As is expected in an ideal vision for nanoscale probing, the electric field is limited to the vicinity of the contact point between the AFM tip and SiC. Inside the SiC sample, an electric field E of around 8.2 × 10^8^ V/m is obtained under the contact point and is divided by a factor 10 at a distance of 20 nm (in depth) away from this point. The presence of the water nanomeniscus (Figure 4b) fundamentally modifies the electric field distribution around the AFM tip and in the SiC layer. The electric field at the SiC surface is lowered by a factor of 10 compared to the results without a water meniscus. It is more homogeneously distributed over the interface area between the absorbed water and 4H-SiC. However, the field is now concentrated at the rim of the water meniscus and the interface with the SiC, leading to an enhanced field at a position, which is not situated under the tip, but given by the extrinsic shape of the water nanomeniscus.

With the presence of an oxide layer, the electric field distributions (at the SiC surface and inside materials) are shown in Figure 4c. An enhancement of the electric field inside the oxide layer and around the edge of the AFM tip is discerned. Again, the location with the highest electric field is the point of the rim of the water meniscus at the sample surface. Profiles of the electric field at the SiC surface, or SiC/oxide interface, in the presence of a water meniscus, are represented in Figure 4d. The electric field is quasi-constant over 40 nm (i.e., water meniscus width) with a value around 1.5 × 10^8^ V/m. The impact of the oxide thickness was also studied, and oxide thicknesses from 3 nm to 6 nm were computed. Results show that the oxide thickness has only a weak influence on the electric field distribution at SiC surface both in air as well as in the water. It is interesting to note that the presence of the SiOCH layer decreases only weakly the electric field at the SiC surface. This implies that the growth of the oxide layer during anodization by the AFM tip slowly decreases the electric field and is consequently a self-limiting growth phenomenon. Far from the meniscus, a strong decrease in the field is calculated. These results also imply that the electric field distribution will be influenced by the shape of the water meniscus (i.e., the variation of the environmental humidity) and the shape of the tip (i.e., tip ageing during contact C-AFM measurements) leading to a variety of anodization area sizes and oxide thicknesses.

### 3.2. Electrical Conduction at the Nanoscale

In order to evaluate the effect of the doping level of the SiC layer on the anodization, different epitaxial layers have been studied. In Figure 5, AFM topographies of the n-doped 4H-SiC sample with different doping layer levels before and after C-AFM measurement are presented. During C-AFM, a bias of −10 V applied to the conductive tip. The scan size is 5 µm × 2.5 µm (with a resolution of 512 × 256 pixels). The evolution of the SIMS profile of the nitrogen doping is superimposed on the AFM mappings as a guide to the eye. After preparation, the cross-sectional sample presents a low surface roughness. Indeed, the average surface roughness Ra and root mean square roughness Rq are equal to 0.5 nm and 0.7 nm, respectively. A surface modification was observed after C-AFM measurement in the highly doped regions (zones 4, 5 and the substrate), as can be seen from the topographical maps taken at the same location before and after the measurement (Figure 5a). Under polarization, the silicon AFM tip could be considered as a cathode and SiC as an anode. This surface modification is related to the direct oxide growth by LAO. A zoom view of 1.2 µm × 1.2 µm is reported in Figure 5b. Where the doping concentration is maximal (zone 5), we observed a change in height of 2.8 nm while an increase in roughness was observed for the regions with lower doping (zone 4 and substrate). It seems likely that the local nature of the interaction between sample and tip leads to a pixel-by-pixel anodization of the surface during the AFM scan and thus an inhomogeneous effect. Where the doping level is lower (zones 1, 2, 3 and 6 in Figure 5a,b), we did not observe a significant modification of the sample surface. This is clearest in zone 6, which becomes easy to distinguish topographically after local anodization. The AFM-LAO intensity correlates strongly and selectively with the SiC doping level as a result of local surface oxidation due to negative tip bias during C-AFM scans.

Figure 6 shows the profiles of the topography along the epilayers before and after AFM-LAO. Before AFM-LAO, the form of the topography is due to the surface preparation. Compared to the topography after the C-AFM scan (after AFM-LAO), the change in height is confirmed for the layers 4, 5 and the SiC substrate. Therefore, the doping level threshold where LAO appears is is 7 × 10^18^ at/cm^3^.

With the C-AFM, the local current can be measured in the spectroscopy mode or during the scanning measurement. Figure 7 reports the measured current during the tip approach and retract for the SiC layer with a doping level of 4 × 10^17^ at/cm^3^. This curve is recorded during the experiment presented in Figure 3b (V_DC_ = −10 V). Far from the sample surface, no current flows from the tip through the back contact of the sample. After the tip’s contact with the sample, at z = 1.9 µm, an increase in the current is measured. When the contact is maintained (i.e., z > 1.9 µm), a current of −2 pA is probed through the tip and the back contact. During the retract step, the collected current follows the force curve and current is collected through the water meniscus, even if the physical contact between the tip and the SiC surface is removed. This signal attests to the collection of the current through the water meniscus during this approach–retract experiment at V_DC_ = −10 V. As a consequence of the high electric field, the local current also participates with the LAO mechanism.

Simultaneously to the recorded topography, the local currents can also be measured. For the 4H-SiC staircase sample, the current profile along the epilayers are plotted in Figure 8. As expected, the current depends on the doping level of the SiC layer. To probe the electrical properties of the grown oxide layer, currents are collected before and after LAO. For the highest doping level (#5), the decrease in the current after LAO confirms the insulating behavior of the grown oxide.

To quantify the influence of the doping level on the anodization and on the measured current by C-AFM, the current density collected by the AFM tip is computed by FEM using Equations (1) and (3). The current density distributions calculated without oxide and with an oxide layer are shown in Figure 9a,b, respectively. The resulting currents collected by the AFM tip for different doping levels are reported in Figure 9c,d. Without the oxide layer, the current increases by a factor of 10 from the lowest to the highest doping level. Moreover, depending on this doping level, two regimes can be identified. At low doping (less than 5 × 10^18^ at/cm^3^), the current only increases slowly. At a high-doping level (higher than 5 × 10^18^ at/cm^3^), the current increases much more drastically. To simulate the effect of the LAO, similar simulations are performed for the presence of oxide layers with three different conductivities (Figure 9d). The oxide presence implies a decrease in the current collected by the AFM tip with increasing SiOCH thickness. This is consistent with the observed decrease in the C-AFM current after anodization, as experimentally observed in Figure 8. Moreover, the FEM approach demonstrates that the influence of the SiOCH thickness is more pronounced for oxides with high conductivity, which indicates low-quality oxides with defects or impurities leading to bad insulating properties.

A dependence of the anodization with the doping level of the SiC was shown experimentally. According to FEM, this anodization occurs for the conductive regime in which the current depends strongly to the doping level. Consequently, anodization and oxide growth are favored by high currents flowing through the tip–sample contact and high electric field applied to the tip. Indeed, a thicker oxide strongly decreases the current flowing through the SiC and the electric field distribution. Moreover, the FEM results demonstrate that the decrease in the current during C-AFM measurement is related to the presence of a poorly insulating SiOCH layer whose conductivity is around 1 S/m.

## 4. Results for the SiC-MOSFET

A commercial SiC-MOSFET is studied, characterized and analyzed. The topography acquired on the cross-sectional sample is reported on Figure 10a. The topography reveals the SiC MOSFET structure at the die level. The gate (G) and the source (S) contact can be recognized. Moreover, around the gate, SiO_2_, inter-metal dielectric layers and aluminum contact for the source can be identified. The mapping of the AFM deflection signal error is also reported in Figure 10b, allowing to better appreciate the structure of the device. 

In Figure 11, the surface topography evolution for successive C-AFM acquisitions is presented. For all measurements, a negative voltage of −10 V is applied to the AFM tip during scanning. By comparing with the acquisition before applying the negative voltage, the AFM acquisition of the topography reveals new areas with a higher topographic contrast. These news areas are the oxide created during the local oxidation of the SiC. Very interestingly, the areas are located on both sides of the MOS channel, at the location of the highly n-type-doped layers (Figure 11a). For the less-doped regions (for example, the n-drift layer), no oxide formation was detected for the used experimental parameters, as expected from the results of the previous section showing the strong dependence on the SiC doping level. For the second (Figure 10b) and the third (Figure 11c) scans with negative bias, the zones showing a local oxidation stay the same, but with an enhancement of the LAO effect. During successive scans, for the LAO oxide growth, two mechanisms are involved: (i) the decreasing of the current density in the water meniscus which decreases OH^−^ transport to the anode and (ii) a diffusion barrier for OH^−^ to reach the SiC surface. This implies that oxide growth is an auto-limiting effect.

In addition, to reveal the nature of the grown oxide by LAO, a selective wet chemical etching is performed on the sample. A highly concentrated solution of potassium hydroxide (KOH, 2M) is used to dissolve SiO_2_. This choice of the etching solution was made so as to slowly etch the pure SiO_2_ layers in order to avoid as much as possible a surface degradation of the sample, and at the same time, preserve the pure SiC zones [43], which will act as reference zones for the evaluation of the height changes. The sample is immerged for around 10 min in the solution at room temperature. Topographies are recorded before and after the etching. For this experiment, in order to improve the resolution of the topography scan, the Scan Asyst mode is used with a AFM tip with a radius of 2 nm. Figure 12a,b show the impact of the KOH etching on the cross-section after LAO, in the areas cobining the anodic oxide layers on the highly doped SiC situated just below the SiO_2_ gate oxide. The comparision of the topography profiles (Figure 12c) along the axes indicated in the respective figures before and after etching highligths a small effect on the SiO_2_ gate oxide, which is more important than on the oxide layer grown by LAO. The fact that the LAO-grown oxide is etched more slowly than pure SiO_2_ is a strong indication that the oxide layer grown by LAO is mainly a SiOCH layer.

## 5. Conclusions

The signature of the presence of a water meniscus on the AFM observation of SiC local electrical properties and the surface anodization was determined by FEM simulations and C-AFM measurements. The evaluation of the electric field and the current density highlights the influence of the water meniscus, but also of the oxide surface layer, as well as the doping level of the SiC. Being strongly dependent on the details of the water meniscus, the influence of extrinsic parameters, for example, the air humidity, is shown, explaining the scattering of reported experimental results. The simulation results are confirmed by AFM measurements, where the influence of the doping level and its threshold value of 7 × 10^18^ at/cm^3^ for anodization was identified on epitaxial SiC staircase samples with gradual doping. For the COTS SiC MOSFETs, the anodization is present only on the highly doped areas of this complex structure, which can therefore be identified by simple topographical measurements after the application of a sufficiently high field through the AFM tip. The auto-limiting effect LAO oxide growth is also characterized by the study of the consecutive scans. However, the SiOCH nature of the oxide grown by surface anodization was demonstrated by selective KOH etching. This investigation highlights the LAO effect for doped SiC and demonstrates that this effect must be taken into account during electrical AFM measurements, in particular, in the case of successive scans This result could also provide a basis for the design and fabrication of AFM-oxidized nanostructures, using doping as an oxidation mask.

## Figures and Tables

**Figure 1 materials-16-05401-f001:**
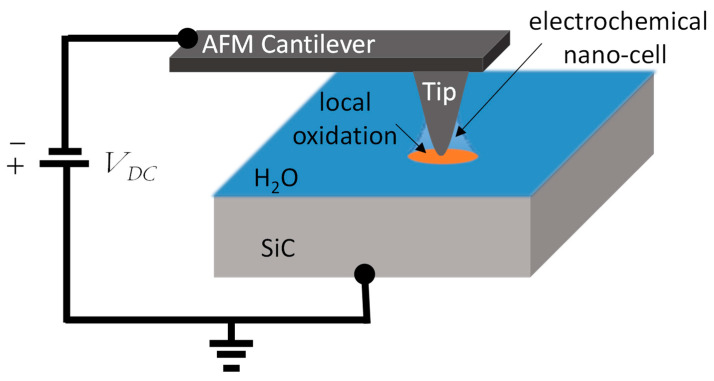
AFM tip in contact with the SiC with the presence of absorbed water: creation of an electrochemical nano-cell by applying a negative bias to the tip and local anodization oxidation.

**Figure 2 materials-16-05401-f002:**
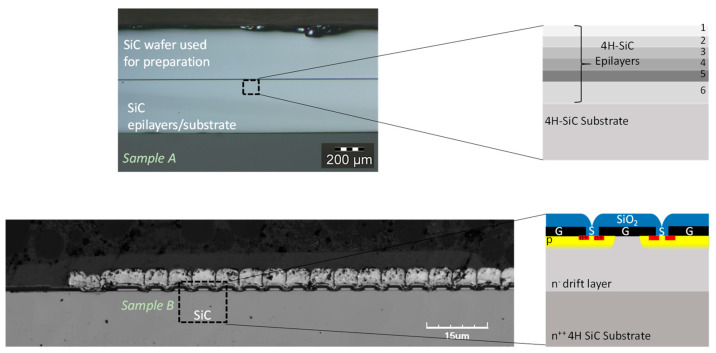
Studied SiC samples: Sample A, optical view of the cross-section of the epitaxial SiC multilayer deposited on the 4H-SiC substrate and Sample B, confocal microscopy view of the cross-section after the surface preparation and a schematic of the classical MOSFET structure.

**Figure 3 materials-16-05401-f003:**
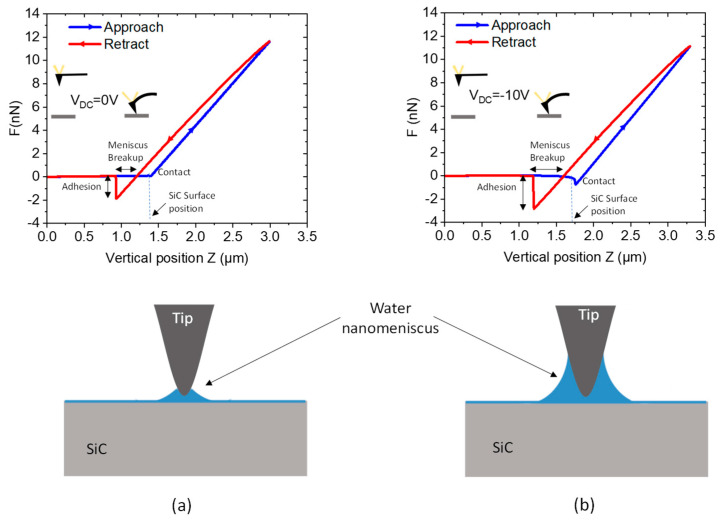
AFM force–distance curves for two applied DC bias: (**a**) V_DC_ = 0 V and (**b**) V_DC_ = −10 V. Approach and retract curves are with schematics of the cantilever position, and schematics of the water nanomeniscus bridging the AFM tip and the sample surfaces for the two polarizations.

**Figure 4 materials-16-05401-f004:**
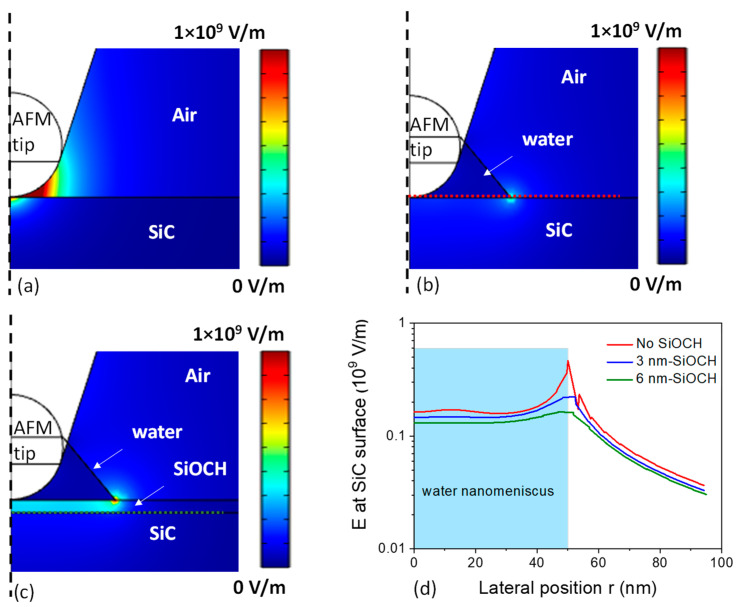
FEM simulation of the electric field distribution (**a**) without water, (**b**) with a water meniscus and (**c**) with a 6 nm thick oxide layer (10 V is applied on the AFM tip). (**d**) Electric field profile along SiC surface for various oxide thickness.

**Figure 5 materials-16-05401-f005:**
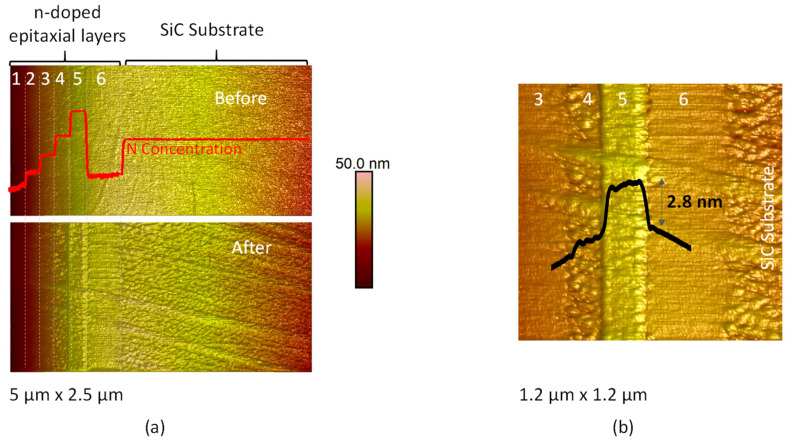
(**a**) AFM topographies (5 µm × 2.5 µm) before and after the AFM-LAO for the SiC-doped staircase sample; the SIMS profile is reported and (**b**) so is a zoom on the grown oxide.

**Figure 6 materials-16-05401-f006:**
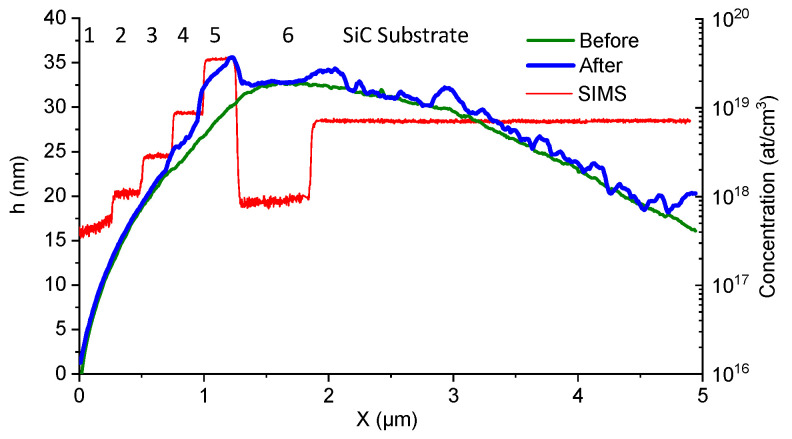
Profiles of the topography along the epilayers before and after AFM-LAO. The red line indicated the N concentration from SIMS technique. The doping layers are identified with numbers from 1 to 6.

**Figure 7 materials-16-05401-f007:**
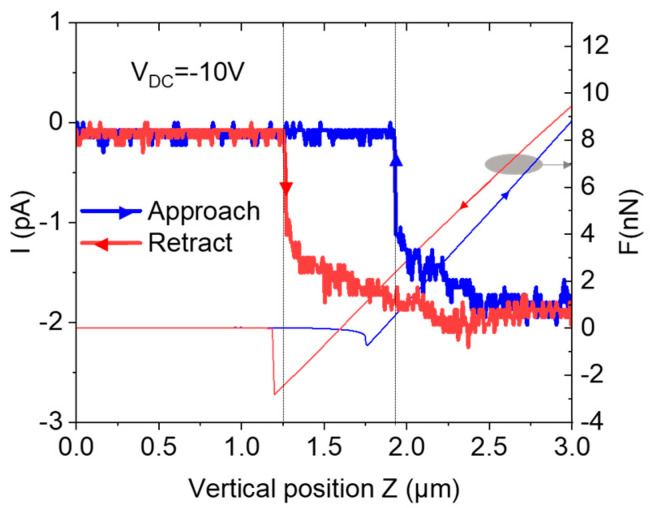
Evolution of the collected C-AFM current during the tip approach and retract with an applied bias voltage V_DC_ = −10 V. The curve forces are superimposed.

**Figure 8 materials-16-05401-f008:**
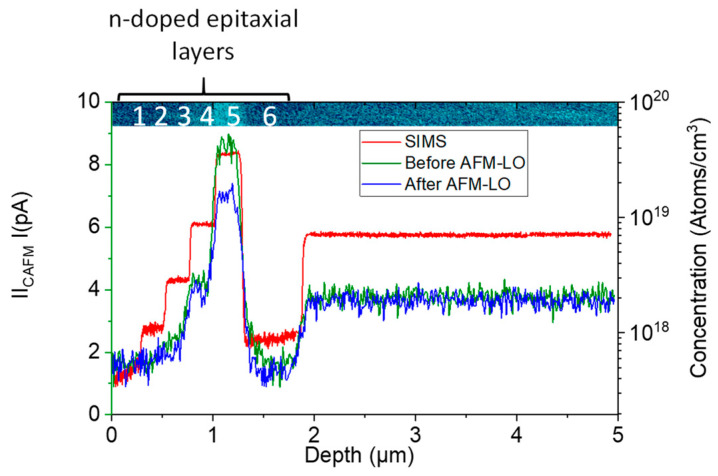
Profiles of the current along the epilayers before and after AFM-LAO. The red line indicated the N concentration from SIMS technique.

**Figure 9 materials-16-05401-f009:**
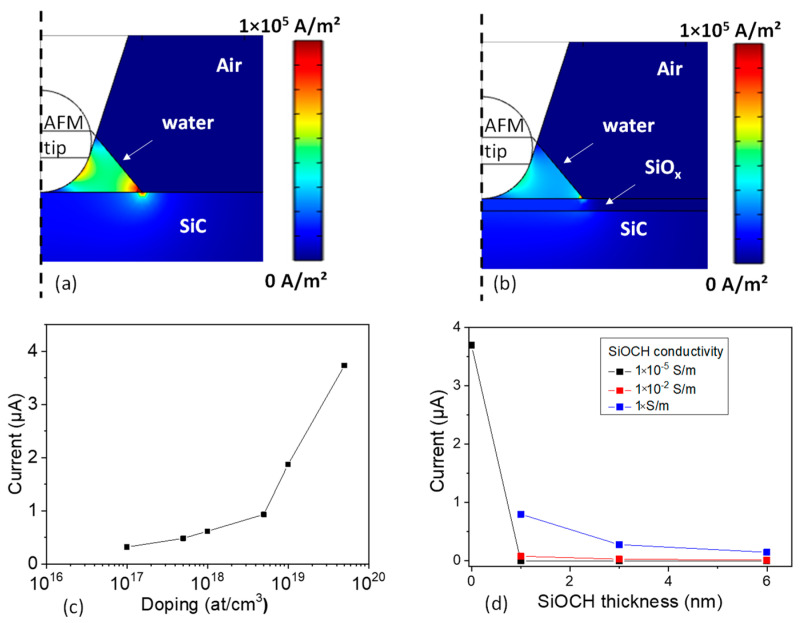
(**a**) FEM simulations of the surface current distribution without the oxide layer and (**b**) as function of SiC doping; −10 V is applied on AFM tip. (**c**) FEM simulation of the surface current distribution with the oxide layer and (**d**) evolution of the current collected by AFM tip as function of oxide thickness and conductivity.

**Figure 10 materials-16-05401-f010:**
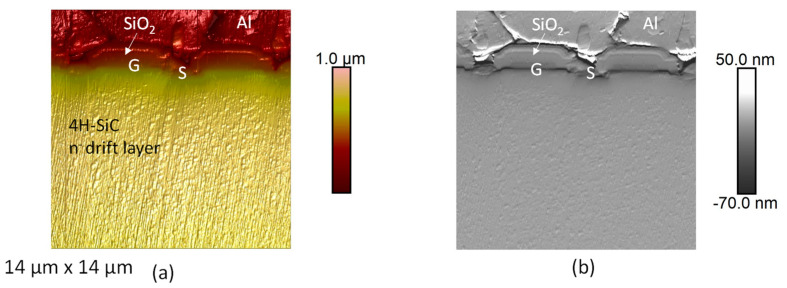
AFM topographies of a commercial SiC-MOSFET (**a**) topography and (**b**) deflection error signal.

**Figure 11 materials-16-05401-f011:**
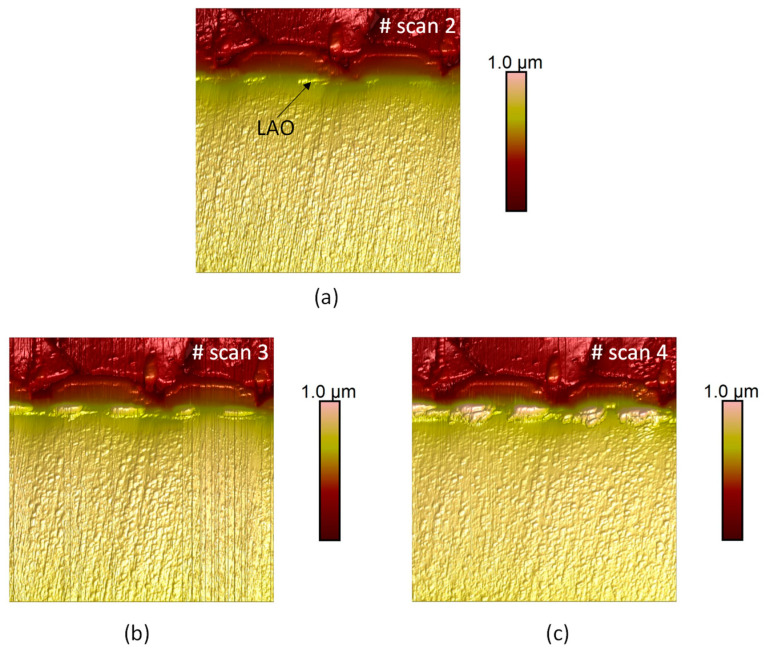
Consecutive AFM topographies of a commercial SiC-MOSFET: (**a**) second, (**b**) third and (**c**) fourth scan with V_DC_ = −10 V applied on the tip. Scan size: 14 µm × 14 µm.

**Figure 12 materials-16-05401-f012:**
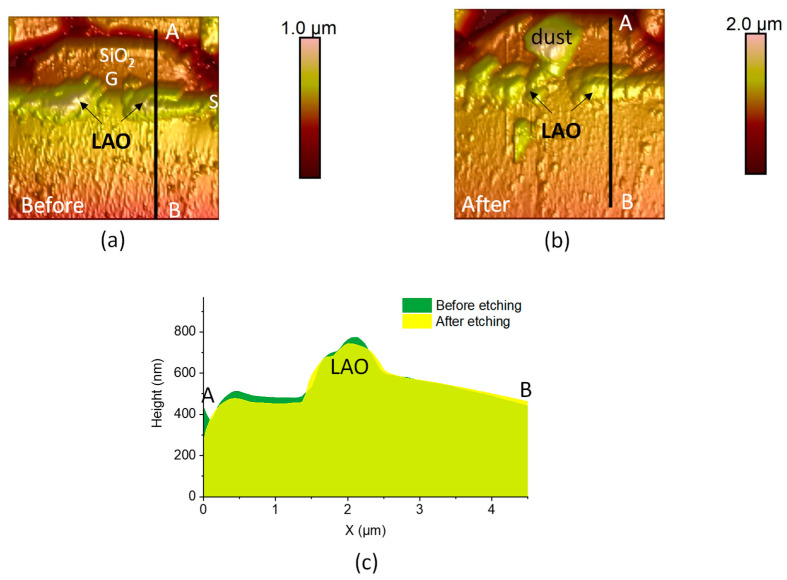
Topographies (**a**) before, (**b**) after the etching and (**c**) comparison of the profiles along the AB line after and before etching. The scan size is 6 µm × 6 µm. S is the source and G the gate of the SiC MOSFET.

## Data Availability

Not applicable.

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
