# Peer review of "SiC Doping Impact during Conducting AFM under Ambient Atmosphere"

_materials, 2023, doi:10.3390/ma16155401_

Round 1

Reviewer 1 Report

The article presents a comprehensive investigation of the anodization effects and local oxidation mechanism of Silicon Carbide (SiC) using specific electrical Atomic Force Microscopy (AFM) modes. The experimental results are supported by an electrostatic modeling approach to understand the oxide growth. Overall, the study is well-executed and provides valuable insights into the behavior of SiC under C-AFM measurements. However, there are some areas that need improvement and clarification before the paper can be considered for publication.

·       The introduction provides a brief overview of the specific electrical AFM modes for the characterization of SiC at the nanoscale. However, the authors should  highlight better the novelty and the importance of their work in comparison to previous research.

·       The conclusion section provides a brief summary of the main findings; however, it lacks a critical reflection on the limitations of the study and suggestions for future research directions. Including these elements would enhance the overall impact of the paper.

·       The authors should include representative AFM topographies showing the localized oxide formation and clearly identify the highly n-type doped zones. Moreover, a discussion on the implications of localized oxidation for SiC-MOSFET performance should be included to provide a comprehensive understanding of the findings.

·       The authors should provide representative AFM images or profiles illustrating the impact of the water nanomeniscus on the SiC surface. Additionally, a quantitative analysis of the AFM data, such as the dependence of the oxide growth on the doping level, should be included to support the experimental findings.

·       The authors should explain better  the physical and chemical processes involved in the anodization and provide a comprehensive explanation of how the water nanomeniscus facilitates the oxidation process. This will enhance the understanding of the readers and strengthen the novelty of the study.

In conclusion, this study has the potential to make a significant contribution to the field. I recommend that the authors address the comments outlined above to improve the manuscript and ensure its suitability for publication.

Moderate English revision 

Author Response

We would like to thank you for your consideration and the review of our manuscript. We also thank the reviewers for the valuable comments which will help to improve this manuscript. In agreement with your suggestions and those of the reviewers, we have updated the manuscript, with new investigations and clarifying a number of points. The corresponding changes in the manuscript are highlighted.

With concern to the reviewers' comments, Please see the attachment.

Reviewer 2 Report

In this work, the signature of the presence of a water meniscus on the AFM observation of SiC local electrical properties and the surface anodization was determined by FEM simulations and C-AFM measurements. The evaluation of the electric field and the current density high- lights the influence of the water meniscus, but also of the oxide surface layer, as well as  the doping level of the SiC. 

Q1: Can the SiO2 oxide and the SiOCH layer induced by  local anodic oxidation be proved by experimental tests? What is the origin of the C element? 

Author Response

We would like to thank you for your consideration and the review of our manuscript. We also thank the reviewers for the valuable comments which will help to improve this manuscript. In agreement with your suggestions and those of the reviewers, we have updated the manuscript, with new investigations and clarifying a number of points. The corresponding changes in the manuscript are highlighted.

With concern to the reviewers' comments, please see the attachment

Reviewer 3 Report

The results are sufficient but may be the contribution of your results should be discussed more

proof read is needed

Author Response

(The authors gave the same response as above.)
